# Model for Interference Evaluation in 5G Millimeter-Wave Ultra-Dense Network with Location-Aware Beamforming

**Grigoriy Fokin *** and **Dmitriy Volgushev**

Software Defined Radio Laboratory, The Bonch-Bruevich Saint Petersburg State University of Telecommunications, 193232 Saint Petersburg, Russia
* Correspondence: grihafokin@gmail.com

**Abstract:** Location-Aware Beamforming (LAB) in Ultra-Dense Networks (UDN) is a breakthrough technology for 5G New Radio (NR) and Beyond 5G (B5G) millimeter wave (mmWave) communication. Directional links with narrow antenna half-power beamwidth (HPBW) and massive multiple-input multiple-output (mMIMO) processing systems allows to increase transmitter and receiver gains and thus facilitates to overcome high path loss in mmWave. Well known problem of pencil beamforming (BF) is in construction of precoding vectors at the transmitter and combining vectors at the receiver during directional link establishing and its maintaining. It is complicated by huge antenna array (AA) size and required channel state information (CSI) exchange, which is time consuming for vehicle user equipment (UE). Knowledge of transmitter and receiver location, UE or gNodeB (gNB), could significantly alleviate directional link establishment and space division multiple access (SDMA) implementation. Background of SDMA is in efficient maintenance of affordable level of interference, and the purpose of this research is in signal-to-interference ratio (SIR) evaluation in various 5G UDN scenarios with LAB. The method, used to evaluate SIR, is link level simulation, and results are obtained from publicly released open-source simulator. Contribution of research includes substantiation of allowable UE density, working with LAB. Practical implications include recommendations on terrestrial and angular separation of two UE in 5G UDN scenarios.

**Keywords:** 5G; UDN; wireless communications networking; location-aware beamforming; positioning; interference; millimeter waves

## 1. Introduction

The rapid growth in the number of simultaneously operating transceivers in emerging and deployed 5G New Radio (NR) and Beyond 5G (B5G) ultra-dense networks (UDN) leads to the problem of an unacceptably high level of interference, provided they are densified to one device per square meter [1,2]. Adaptive beamforming (BF) can potentially compensate high interference level by maximizing the antenna radiation pattern (ARP) [3–5] to the source/receiver of the desired signal and minimizing the antenna pattern to the source/receiver of the interfering signal [6,7]. Widespread use of beamforming in 5G is facilitated by the transition of the radio interface to millimeter wave (mmWave) [8,9] and the development of massive multiple-input multiple-output (mMIMO) antenna systems [10,11]. Directional links with pencil beams at gNodeB (gNB) and user equipment (UE) alleviates path loss by high antenna array (AA) gain at the transmitter and receiver [12,13]. Minimization of intra-system interference of devices, operating in common frequency range in beamforming mode, is carried out due to their spatial multiplexing and angular/terrestrial separation [14,15]. In particular, half-power beamwidth (HPBW) influence evaluation on inter-beam interference in mmWave 5G NR UDN with uniform rectangular array (URA) gNodeB (gNB) in [15] shows, that azimuth beam narrowing from 60° to 10° results in approximately 15-dB interference suppression for hexagonal layout.

To control ARP the well-known approach of preliminary analysis of training sequences about the current situation in the channel with CSI (Channel State Information) can be

used [9], but its overhead become unacceptably high in UDN for vehicle user equipment (VUE) [16]. An alternative approach to beamforming (BF) is the so-called Location Aware Beamforming (LAB) [17]. The validity, relevance and perspective of this approach is determined by the fact, that for 5G UDN in the latest 3GPP specifications, in contrast to the networks of previous generations [18], the requirements for positioning accuracy of user equipment (UE) up to one meter were formalized for the first time [19].

A survey of enabling technologies for network localization, tracking, and navigation in [20] highlights shift from positioning in two dimensions to three-dimensional location estimation in 5G UDN environments using range-free localization schemes. Overview of positioning architectures in previous generations of cellular networks and prospective positioning architecture and technologies in [21] discuss potential in achieving sub-meter accuracy in 5G networks. Achieving this accuracy became possible due to new radio interface at the physical layer [22–24], however, the new device-centric architecture of the UDN, including direct communication with each other in D2D (Device-to-Device) mode, creates new problems of beam management at the link and network layers [25–27].

Initial research in the field of LAB has already been carried out over the past years, but mainly for a particular scenario of one radio link between a base station and a UE in order to estimate UE location [28–31] and for a particular scenario of the mutual influence of two links between one base station and two fixed UE for the purpose of interference evaluation [32–38]. The problem of evaluating interference in radio links with beamforming for vehicles is complicated by following factors: significant (tens of dB) dependence of the instantaneous signal-to-noise-plus interference ratio (SINR) on the current spatial [1] and angular [16] separation of the VUE; interdependence of channel coherence time and beamwidth in vehicular channels [39–41]; dependence of beamformed radio link capacity on UE positioning uncertainty [42]; employed beam tracking approaches [43,44]; beam alignment techniques [45–48] and location-aided channel estimation [49–53].

Recent investigations [16–22], concerning LAB in 5G UDN for mmWave links, treated only stationary gNB and UE cases. The aim of current research is to evaluate interference level in 5G mmWave UDN scenarios with Location-Aware Beamforming, when gNB perfectly knows VUE coordinates and points its pencil beam directly to VUE during its movement in various scenarios, which are characterized by terrestrial and angular separation.

One of the first works, concerning LAB for fixed UE [54], revealed, that using position enables in creating transmit precoding BF vector for line-of-sight (LOS) scenarios without signaling overhead. In particular, proposed approach was to limit beamforming to spatial area, bounded by angular sector with inaccuracy of the GPS positioning up to 20 m.

Fundamental purpose of the project "Location aware beamforming in mm-wave band ultra-dense radio access networks" [55] is to develop scientifically grounded methodology for ARP control, based, on the positioning of UE for scenarios of a separate link, two links, a set of links of one cell and a set of UDN cells.

To numerically substantiate the terrestrial and angular separation of neighboring transceivers in a given set of conditions for networks development and scenarios for the device operation, signal-to-interference ratio (SIR) evaluation should be taken into account, which will make it possible to scientifically substantiate the theoretical and practically realizable limits of separation for simultaneously operating in the spatial multiplexing mode transceiver devices with LAB in 5G and B5G UDN.

Motivated by the described state-of-art, this work aims to evaluate interference in 5G mmWave UDN with LAB for scenarios of two links.

The paper is organized in the following order. Section 2 describes background for location-aware beamforming in 5G ultra-dense networks. Section 3 presents publicly released open-source simulator for LAB in 5G UDN and performs interference evaluation through link level simulation. Finally, paper is concluded in Section 4.

## 2. Background for Location-Aware Beamforming in 5G Ultra-Dense Networks

5G and B5G UDN in the current decade are expected to continue densification trend with the increase of base station or gNB and UE number per unit area in 2D and unit volume in 3D. However, network densification has fundamental limits which can be explained by the following analogy [56]: area spectral efficiency gain reaches saturation, when the number of wireless transceiver devices and spectrum reuse increase, likewise, semiconductor industry reaches a plateau of Moore's law, when chips computing power is no longer doubled every two years. For example, in [56] it is reported, that in the interference limited scenario network densification initially leads to SINR increase, however, after some finite gNB density is reached, potential throughput reach limit and can even decrease to zero, if pathloss exponent is below 2 and 3 for 2D and 3D cases respectively. Directional transmissions for multi-user (MU) mMIMO in mmWave are seen as one of the solutions to push fundamental densification limits as far as possible.

Analysis of radio-frequency pollution (RFP) for UDN scenarios in [57] proves, that densification of gNB, working in omnidirectional mode, does not lead to exponential increase of RFP. As for gNB in multi-user BF mode, it supposes, that perceived by UE RFP level should be substantially lower, than theoretical maximum with omni gNB, because beams are usually activated by gNB to temporarily cover UE locations. Analytical approach in [57] appears to be capable for estimation not only RFP levels, but also upper and lower bounds of SINR, which varies both in time and space for UDN with gNB and UE, working in BF mode.

Assessment of 5G network with mMIMO in [58] states, that spatial distribution of UE, served by gNB in BF mode, largely impact electromagnetic field (EMF) exposure in both azimuth $\varphi$ and elevation $\theta$ dimensions, and maximum power level is obtained for UE with focused beams. To tract analytically the uncertainty of UE spatial distribution, authors in [58] propose four scenarios of directional link establishment, defined by the probability, that UE at each time instant is positioned in the predefined $\varphi$ and $\theta$ direction. For investigation of the dependence of signal power levels on spatial user location authors in [58] introduce time averaging across beam orientations through predefined directions across AA broadside. Main conclusion in [58] is, that time-averaged maximum power levels in BF mode allows to reduce Inter-Site Distance (ISD) between gNB from the EMF exposure point of view and thus facilitates network densification. In spite of some analytical tractability, main drawback of approach in [58] is its inability to assess instantaneous power levels for mobile UE.

One of the first scientific approach to assess pencil BF, using UE location knowledge, is presented in [59] and is supplied with an open-source simulator. The contribution of [59] is a LAB technique, which implements 3D traffic beam steering in azimuth and elevation planes according to UE position estimate with predefined uncertainty level. Besides dispelling the myth about increasing EMF exposure with pencil BF, it provides comprehensive mathematical and simulation model for LAB on the network level of abstraction with intra-sector and inter-sector interference evaluation, depending on the direction and HPBW toward each UE, tuned by gNB. SINR evaluation for multiple-beam system is performed according to approach in [60]. Despite pioneering contribution in LAB field of research, investigation [59] lacks accounting of the following factors: (a) traffic beams adaptation to UE mobility cases; and (b) trade-off between the time needed to adaptively steer the beam and positioning estimate relevance, which needs to be frequently updated for high-speed VUE.

Approach of location-aware beamforming stems from challenges in alternative beam management (BM) procedures in mmWave UDN [61]. First of all, higher carrier frequencies in frequency range 2 (FR2) between 24.25 and 52.6 GHz, and extended from 52.6 to 71 GHz, dedicated to 5G networks, are subject to strong attenuation, thus, in order to achieve coverage, mmWave devices require to use narrower beams with higher AA gain. Narrowing HPBW, in turn, complicates BM procedures during beam sweeping on the initial access [25] and further beam refinement [26] stages. The reason is that narrower beams need larger

transceiver codebook size, causing increase of beam sweeping overhead, complexity and latency [61]. One of the recommendations to accommodate a large number of pencil beams for sparse mmWave MIMO channel (with much less beams, than the number of antenna array elements), is to reduce beam search space, utilizing distributed compressed sensing approach [62]. One more solution for codebook design is to use narrower beam search space using prompt estimates of UE direction of arrival (DOA) and direction of departure (DOD) [27]. Another challenge of BM concerns beam alignment for highly mobile VUE and necessitates tradeoff between timing for beam sweeping, expiration time of beam correspondence state and antenna array HPBW. One of the solutions to this problem is also LAB, relying on predictable VUE trajectories.

To the best of our knowledge, one of the first experimental demonstrator of location-aided beam alignment technology in mmWave band through field trials is described in [63]. Sufficient link margin, needed to overcome severe pathloss in mmWave, can be provided by high gain directional antennas, which necessitates both transmitter (TX) and receiver (RX) to use beam steering with narrow beamwidth. This challenge facilitates development of effective BM procedures and solutions with low latency and signaling overhead, actual for high-speed trains (HST) [64].

Movement assessment in linear and angular directions with inertial measurement units (IMU) for slow vehicles, utilizing BF in 60 GHz links, is evaluated in [65]. In particular, experiment in $10 \times 10 \times 4$ m room with IEEE 802.11ad stations, equipped with $30°$ HPBW antenna, validated possibility to quickly realign beams during 10 ms interval, based on IMU prediction, thus substantiating LAB for low-speed vehicles.

Authors in [63] implemented LAB testbed at 28 GHz with baseband processing in National Instruments (NI) software-defined radio (SDR) universal software radio peripheral (USRP) board NI USRP-2954R, passband processing in Analog Devices (ADI) radio frequency (RF) modules ADF5356, ADMV1013, ADMV4801 and beamforming in uniform planar array (UPA) antenna, comprising $4 \times 16$ patch antenna elements (AE). The beam-steering is based on analog BF approach through the delay-sum beam codebook, capable of scanning azimuth plane from $-40°$ to $40°$ with $2°$ step. Experiment was performed for LOS scenario with single TX-RX pair, separated by 4.2 m, in anechoic chamber when TX is stationary and RX is linearly moving at speed of 0.1 km/h. LAB realization included following steps: (a) estimation of RX location by means of ultra-wideband (UWB) ranging module; (b) angle of departure (AOD) and angle of arrival (AOA) calculation, based on TX and RX positions; (c) steering TX and RX beams according to calculated AOD and AOA respectively. Described in [63] demo accounted beam tracking interval lower than 200 ms, and proved, that LAB enables tradeoff between beam alignment accuracy and timing even for large-scale AA with pencil HPBW at both TX and RX. However, for high speed VUE in outdoor scenario, using 3GPP network positioning trilateration technologies, LAB demonstrator had not yet been tested.

One of the first experimental investigation of outdoor mmWave propagation, using small form factor steerable antennas for mobile UE, to the best of authors knowledge, was published in [66]. Field trials for 38 GHz radio propagation in urban scenario revealed, that transmitters, elevated at heights from two to eight stories, require $60°$ scanning range (up to $±30°$ off-boresight) in the azimuth, while receivers would benefit from larger azimuth scan range. Study [66] indicates, that steerable antenna architectures at mmWave should work best in dense urban environment with ISD up to 200 m, where non-line-of-sight (NLOS) are rarely preferred to LOS or partially obstructed LOS links, since NLOS have from 10 to 50 dB higher pathloss. Also [66] forecasts usefulness of site-specific ray-tracing models due to environment dependency of TX/RX AOD/AOA distribution.

Work [67], devoted to full-dimension MIMO (FD-MIMO) with antenna array elements, placed in horizontal and vertical domains, draws attention to the additional degree of freedom in beam steering, and the need for accounting not only azimuth, but also elevation direction in three-dimensional (3D) spatial channel models (SCM). In particular, it considers

simulation in 3D SCM azimuth, elevation AOD $\varphi_{k,AOD}$, $\theta_{k,AOD}$ and azimuth, elevation AOA $\varphi_{k,AOA}$, $\theta_{k,AOA}$ for each multipath component (MPC) $k = 1, \ldots, K$.

Investigation [68] shifts focus from point-to-point MIMO links to the state-of-the-art signal processing and information theoretic overview of MU mMIMO with large-scale antenna systems, consisting of dozens and more AEs, being typically significantly larger, than the number of UE, and thus providing additional degrees of freedom for simultaneous individual beamsteering for signal-of-interest (SOI) and null interference for signal-not-of-interest (SNOI). It also presents one example about feasibility of CSI acquisition in frequency division duplex (FDD) systems: channel coherence interval of 1 ms $\times$ 100 kHz in time-frequency grid can support transmission of 100 orthogonal pilot waveforms, and if there are 100 AE at the gNB, the whole coherence interval should be used only for training, leaving no symbols for data transmission. One of the solutions to mitigate non-orthogonal pilot contamination is to use AOA-based methods for each UE. This circumstance serves as an additional justification for the use of LAB instead of traditional CSI acquisition both in uplink (UL) and downlink (DL), especially in LOS conditions.

Three-dimensional beamforming (3DBF) [69] promises to substantially improve capability of intracell interference management, comparing with only two-dimensional (2DBF) beamforming. 3DBF allows per user beam tracking due to electronically adapting ARP orientation both in azimuth and elevation planes according to specific UE current location. Applying additional coordination between neighboring cells, it becomes possible to also alleviate intercell interference. Monte Carlo simulations of dynamic 3DBF [69], when vertical beam pattern is adapted according to UE location, results in 109% throughput gain over the 2DBF at the cell edge with signal-to-noise ratio (SNR) is 0 dB; elevation HPBW decrease from $24°$ to $8°$ doubles throughput when the cell edge SNR is 10 dB.

Experimental prototype of mmWave adaptive BF algorithm realization is described in [70]. It uses 28 GHz carrier frequency, 500 MHz bandwidth and hybrid BF scheme: 32 AEs comprise UPA with 8 horizontal and 4 vertical elements; 8 antenna elements are grouped into a sub-array, thus requiring only 4 RF units per analog channel with a set of phase shifters to form a desired beam pattern. Resulting HPBW is approximately $10°$ in azimuth and $20°$ in elevation plane with BF gain of 18 dBi. To reduce CSI exchange, codebook contains a set of predefined overlapping beams, each with a unique identifier, further supplied to baseband modem. Developed prototype performed beam alignment within 45 ms.

Work [71] points out challenges of spatial multiplexing processing, inherent to massive MIMO and concerning CSI acquisition in both UL and DL. If in UL of 5G NR measuring pilots, transmitted by each UE, can be accomplished by gNB, acquisition of CSI in DL is more complicated. Long-Term Evolution (LTE) standard performed channel estimation for conventional MIMO in DL in the following order: eNodeB (eNB) transmitted pilots, UE measured them and returned back CSI estimate. However, what was feasible in the networks of the previous generation LTE for conventional MIMO, is a challenge in 5G NR for massive MIMO due to the following circumstances [71]. Orthogonal pilots for mMIMO should occupy huge time-frequency resource in DL, proportional to the number of gNB AE, which is an order of magnitude greater than in eNB with conventional MIMO. Consequently, for FDD mode, UE also should occupy huge time-frequency resource in UL to feedback CSI estimate to gNB. If we talk about time division duplex (TDD) mode, problem can be alleviated due to reciprocity between the UL and DL, however it is still a challenge for scenarios with high mobility. Authors in [71] among research problems for mMIMO distinguish so called "Non-CSI@TX operation:", which means, that before a link with UE has been established, gNB has no way of knowing CSI. At the same time, one of the ways to solve this problem could be LAB approach.

Comparison between beamforming and spatial multiplexing (SM) in mmWave MIMO wireless communication systems is performed in [72] and show, that SM and BF could work in tandem. However, SM is more effective in predominantly NLOS links with sufficient channel diversity or rank to support multiple parallel streams in bandwidth-limited high

SNR regime, while BF is more effective in predominantly LOS links with inherent to mmWave high pathloss (PL) in power-limited low SNR regime. Measurement campaigns for 28 and 73 GHz revealed, that beams, properly aligned at both transmitter and receiver sides, are characterized with minimum number of MPC, root mean square (RMS) delay spread and propagation PL; this in turn, facilitates exploiting narrow beamwidth and single-carrier modulation without equalization. Among beam training authors in [72] distinguishes codebook-based, AOD based and long-term BF approaches, and each of these methods can be improved by knowing UE/gNB location in advance, so that to decrease beam space and speed up beam search during initial beam sweeping.

Work [73] addresses implementation issues of large-scale antenna systems (LSAS) in mmWave in general, and hybrid BF schemes in particular. Hybrid BF structure includes a much lower number of digital transceivers, than the total number of AE, thus each transceiver is connected to multiple AE, and the phase of AE is controlled through a network of analog phase shifters. Authors in [73] investigate optimal ratio between the number of transceivers and corresponding number of AE per each transceiver and seek reference signal design to obtain DL CSI, taking an assumption, that partial information about AOD and AOA is available. Simulation results of 32-element uniform linear array (ULA) revealed, that AOD estimation accuracy of RS with hybrid BF is limited by the main beam direction of analog BF on each transceiver. In particular [73], the further main beam direction of hybrid BF is away from the analog BF main beam direction, the smaller is the gain.

Presented background reveals several problems in organizing directional links in 5G UDN and substantiates location-aware beamforming as a solution. Consider further briefly approaches for interference evaluation in 5G mmWave UDN.

Investigation [74] consider deployment of mmWave gNB microcells in the areas with high UE density in addition to traditional sub-6 GHz macrocells. While sub-6 GHz Base Stations (BS) and UE are equipped with omni-directional antennas, mmWave gNB utilize directional transmission. Using stochastic geometry and assumption of UE distribution according to Poisson cluster process, authors in [74] develop comprehensive system level analytical framework and analyze various performance metrics, including interference characterization. Monte Carlo simulation validate analytical results and prove significant network performance improvement with deployment of mmWave gNB with directional beamforming. However, on the link level authors in [74] consider directional transmission with model of sectored antenna pattern. According to such model antenna gain can take only two values: $g_s$, if the UE is within gNB main beamwidth and $g'_s$, if the UE is outside gNB main beamwidth, where $g_s$ and $g'_s$ denote the gains of the main lobe and side lobe respectively. Interference characterization would be more accurate if we account complex radiation pattern for antenna array types, depending on the number of array elements.

More realistic approach for interference evaluation in 5G mmWave UDN is reported in [32–38], however, for stationary UE case. To the best of our knowledge, the problem of interference evaluation for the case of mobile UE had not yet been thoroughly investigated. In this work we consider the problem of interference evaluation for the case of two gNB-UE links with mobile UE, as a basis for further evaluation on the level of a set of links inside single UDN microcell and finally, on the network level of a set of UDN microcells. Despite the fact, that considered case of interference evaluation includes only two gNB-UE links, it is rather challenging. First challenge consists in the instantaneous dependency of signal to interference ratio on UE angular and spatial separation, which changes during their mutual motion. Second challenge includes strong SIR dependence on the nonlinear ARP, defined by the AA type and the number of array elements. Third challenge is due to the need to take into account UE positioning accuracy for gNB, utilizing location-aware beamforming. Oversimplification of these factors can lead to the network SIR overestimation or underestimation, which can take values of several tens of dB.

The novelty of current work is in instantaneous SIR evaluation model for cases of two links between two mobile UE and two stationary gNB, equipped with smart antennas,

which perform location-aware beamforming during UE motion in predefined scenarios, accounting for their SOI and SNOI roles. Developed and publicly released open-source simulator includes thorough account of UE mobility and complex models for ARP for various AA types and the number of array elements. Combination of various primary measurements of heterogeneous radio access network for UE location estimation and further beamforming in the simulated scenarios can be accounted with positioning inaccuracy, defined by standard deviation of UE coordinates calculation. Further simulation results are performed for the case of perfect positioning accuracy, and even for such scenario it demonstrates considerable SIR fluctuation, that needs to be accounted for, when assessing UE terrestrial and angular separation. In the further subsection we will consider simulation model of two gNB-UE radio links.

## 3. Link Level Simulation Model for Interference Evaluation in 5G UDN with LAB

The distinguishing feature of the proposed simulator is that gNB knows current UE location and both UE moves along predefined trajectories, which are set by the parameters of terrestrial and angular separation. Such approach allows to scientifically substantiate practically realizable limits of UE SDMA. Section 3.1 considers SIR dependence evaluation on terrestrial separation and Section 3.2 on angular separation of two gNB-UE links. Section 3.3 describes briefly transition from the link to system level simulation model for interference evaluation in 5G UDN with LAB.

It should be noted, that general structure of presented link level simulation model was already detailed in [1]. However, investigation [1] utilized built in functions of MathWorks Phased Array System Toolbox [75]. Presented in the current work open-source simulator utilizes only primitive MathWorks functions and is available at [76].

### 3.1. SIR Dependence on the Terrestrial Separation of Two gNB-UE Links in 5G UDN with LAB
3.1.1. Simulation Model for SIR Dependence on the Terrestrial Separation

The model is designed to estimate the instantaneous signal-to-interference ratio for the scenario of operation of two gNB-UE radio links. Two gNB are stationary and two UE are mobile; gNB is equipped with AA and UE can transmit/receive in either directional (with AA), or omnidirectional (without AA) mode. Supported AA include uniform linear array (ULA), uniform rectangular or planar array (URA) and uniform circular array (UCA). Source code for simulation model is available at [76]. Simulation model consists of three main parts: setting initial parameters and creating scenario, calculating SIR and displaying results; it also has the ability to visualize instantaneous SIR for moving UE.

Initial simulation model parameters include: operating carrier frequency $f$, AA type (ULA/URA/UCA), flag to enable/disable AA at the UE. Each type of antenna array is described by the following parameters: the number of AE $N$, the distance between AE is $\lambda/2$ and is chosen to be equal to the half wavelength $\lambda = c/f$; $c$ is speed of light. To create an AA at the gNB and UE, the createAnt (All functions, comprising open-source simulator, are available at https://github.com/grihafokin/LAB_link_level (accessed on 8 November 2022)) function is used, which forms an array of coordinates $[x, y, z]$ for each antenna array element. These coordinates define the position of the AE in the local coordinate system (CS) of the antenna array, relative to the center of symmetry of the AA. Stationary gNB and mobile UE can use same type of AA, however, gNB in 5G networks are usually equipped with larger AA, compared with UE.

Simulation scenario assumes the presence of two simultaneously operating gNB→UE DL radio links. Each UE works with one gNB during simulation, the selection of the gNB-UE pair is done during creating the calculation scenario. Simulation script determines the location of the gNB and the trajectory of the UE. The gNB parameters, in addition to the coordinates $[x, y, z]$, also include the orientation of its AA. The createNB function is used to initialize the structure of gNB parameters. The UE movement trajectory is specified by the getTrajectory function with following parameters: the starting and ending points of the UE trajectory, the movement speed and measurement time interval.

If there is an AA on the UE, its orientation is also initialized. The createUE function is used to form the UE parameter structure. Each scenario generates a set of straight-line trajectories for the UE. Scenarios under consideration describe urban micro-cell with two gNB ISD = 100 m, and UE moves along street with length of 150 m and width of 10 m. The terrestrial separation of the UE linear trajectories, parallel to the x-axis, are depicted in Figure 1 and are determined by the value $\Delta y$ along y-axis, varying from 0 m to 10 m (street width) with predefined step 0.1 m.

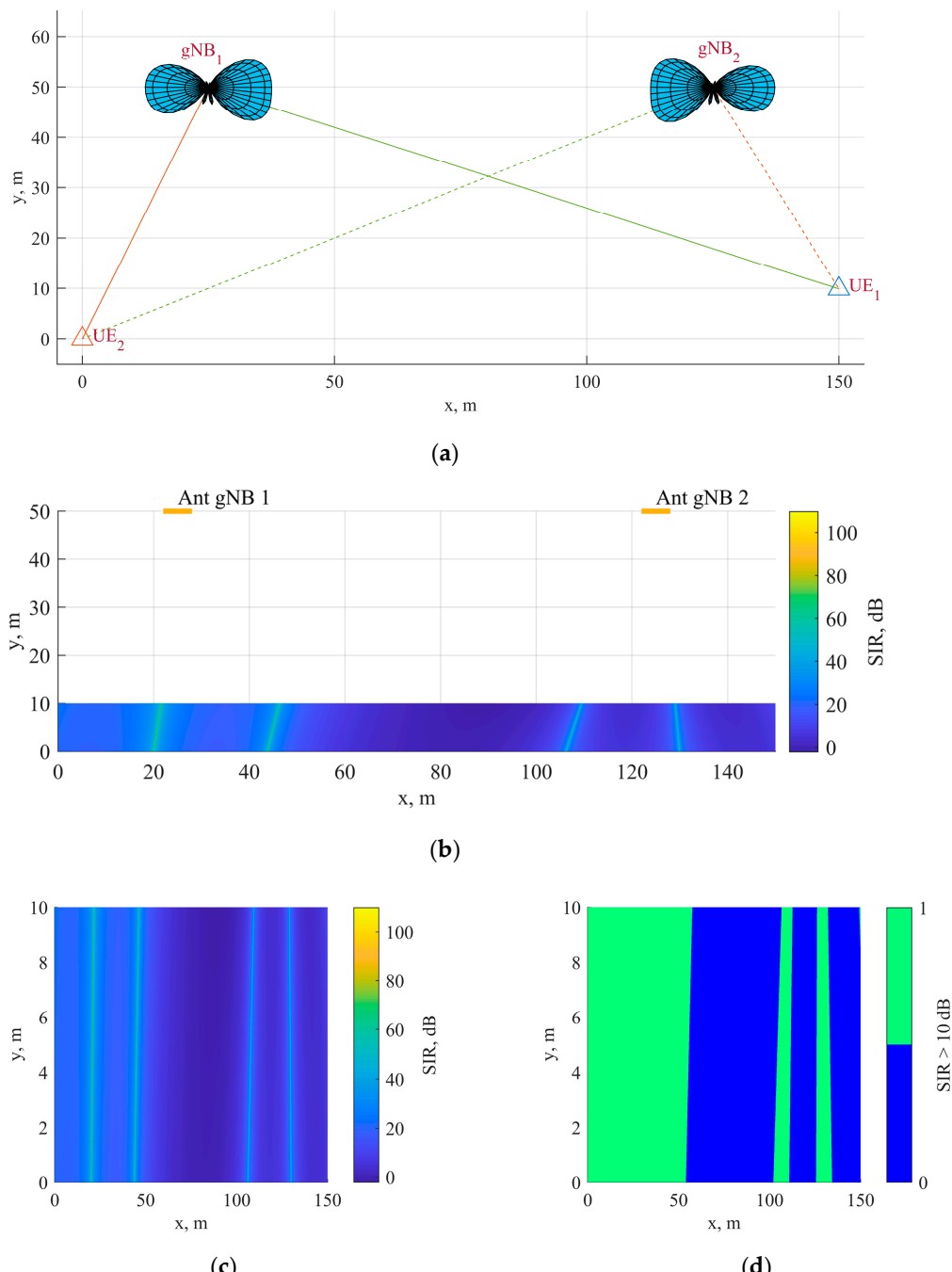

**Figure 1.** Simulation scenario 1: (**a**) gNB with AA pattern and UE trajectories on the x-y plane; (**b**) Map of UE SIR values with gNB location on the x-y plane; (**c**) Map of SIR values along UE trajectories; (**d**) Map of SIR values exceeding a given threshold along UE trajectories.

The model presents two simulation scenarios for two gNB-UE radio links: (1) two UEs moves towards each other at two parallel lines, separated by the distance $\Delta y$, two gNBs are

located on the same road side, relative to UE's trajectories; scenario is in Figure 1a; (2) two UEs moves towards each other at two parallel lines, separated by the distance $\Delta y$, two gNBs are located on opposite road sides of UE's trajectories; scenario is in Figure 2a; green straight line indicates SOI link, while red straight line indicates SNOI link for gNB$_1$; green dotted line indicates SOI link, while red dotted line indicates SNOI link for gNB$_2$.

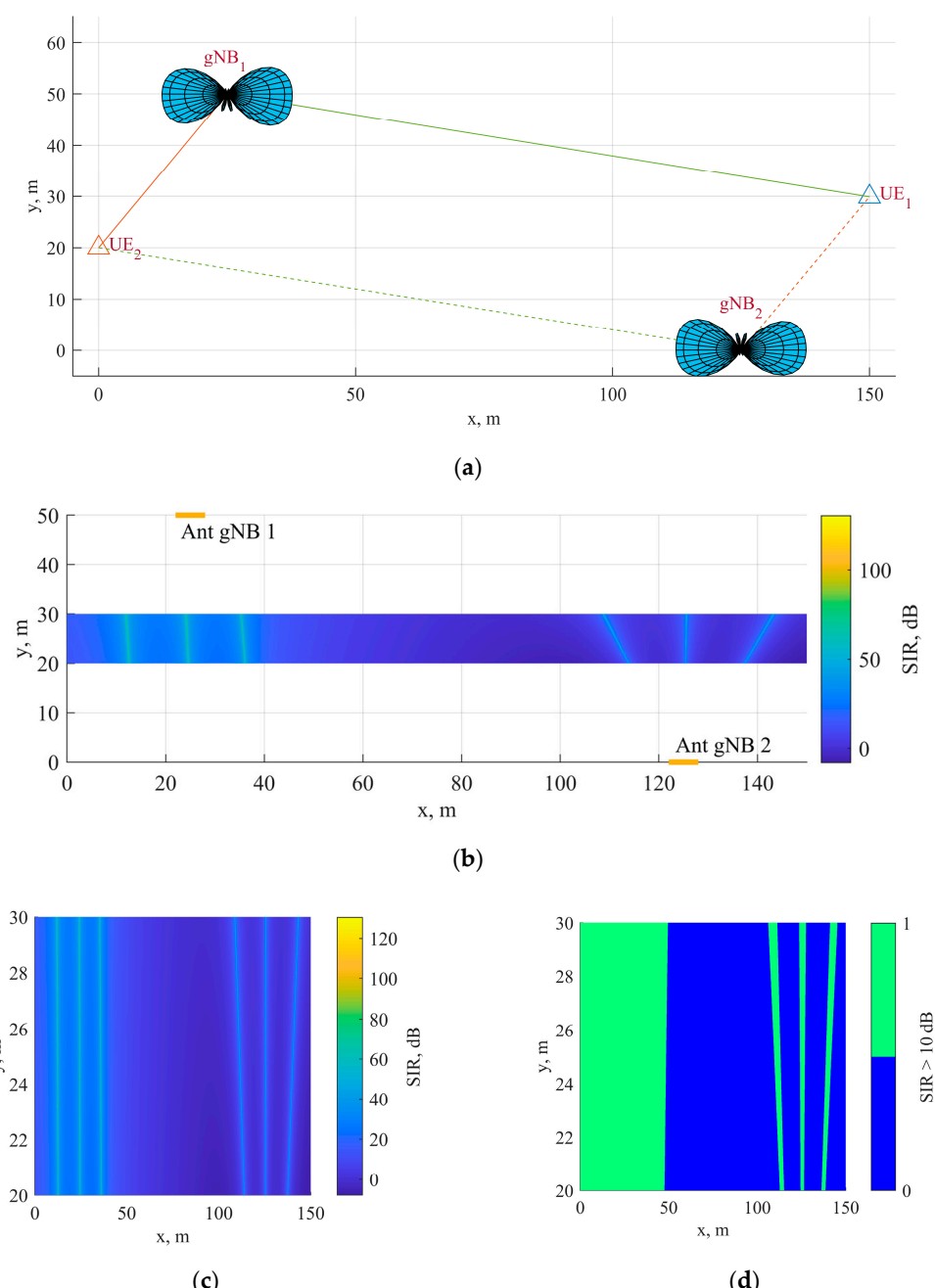

**Figure 2.** Simulation scenario 2: (**a**) gNB with AA pattern and UE trajectories on the x-y plane; (**b**) Map of UE SIR values with gNB location on the x-y plane; (**c**) Map of SIR values along UE trajectories; (**d**) Map of SIR values exceeding a given threshold along UE trajectories.

Figures 1a and 2a illustrates gNB with AA pattern and UE trajectories on the x-y plane. Figures 1b and 2b illustrates a map of SIR values with location and orientation of AA of the serving gNB (Ant gNB 1) and neighboring gNB (Ant gNB 2). Figures 1c and 2c illustrates a map of SIR values in dB; points on the UE trajectory are plotted along the x axis, and the value of *d* is along y axis. Figures 1d and 2d illustrates a map of SIR values, that exceed

a given threshold 10 dB: areas of the SIR map with values, exceeding 10 dB are shaded in green.

Terrestrial and angular separation are not the only aspects, influencing the maximum allowable UE separation. From the physical layer perspective other aspects, such as utilized modulation and coding schemes at the gNB and UE, also contribute to the admissible SINR. However, from the perspective of link and network layer of abstraction, resulting map with SIR values, exceeding a given threshold, could serve as a reference point to assess allowable UE separation in terms of interference distance at first approximation.

From Figures 1d and 2d we can conclude, that at some point, defined by interference distance, $UE_1$, being SOI, begin to experience SIR deterioration, when ARP from SOI $gNB_1$ and SNOI $gNB_2$ are directed to closely located $UE_1$ and $UE_2$ respectively. To overcome such SIR deterioration gNB inter-site distance (ISD) could be reduced. Besides interference distance, authors in [56] investigate BS density per $km^2$ and $km^3$ for allowable network densification evaluation and state, that directional transmissions are seen as one of the solutions to push fundamental densification limits as far as possible.

### 3.1.2. Mathematical Model for SIR Dependence on the Terrestrial Separation

The calculation of the SIR in DL transmission channel at the UE reception side is performed for each point of the UE's trajectory. SIR is calculated, as the ratio of the received power from the serving gNB (SOI) to the power, received from the neighbor gNB (SNOI), expressed in dB. Calculated SIR includes the values of the AA gains of the serving and neighboring gNB towards the UE, depending on the AOD, as well as the free space path loss (FSPL), depending on the distance to the UE. The transmit powers of each gNB are considered to be the same. To calculate gNB→UE AOD, the coordinate systems of the UE and gNB antenna arrays need to be matched. The vector, specifying gNB→UE direction in the global CS, is calculated as:

$$k = x_{UE} - x_{gNB} = [k_x, k_y, k_z]^T, \tag{1}$$

where $x_{UE}$ is the $[x, y, z]$ UE coordinate vector for the current calculation point; $x_{gNB}$ is the $[x, y, z]$ gNB coordinate vector. Translation of the vector $k$ into the gNB antenna array local CS is performed as:

$$r(\varphi_r, \theta_r) = R_{gNB}(\varphi_r, \theta_r)k, \tag{2}$$

$$R_{gNB}(\varphi_r, \theta_r) = R_{gNBz}(\varphi_r)R_{gNBy}(\theta_r), \tag{3}$$

$$R_{gNBz}(\varphi) = \begin{bmatrix} \cos(\varphi) & -\sin(\varphi) & 0 \\ \sin(\varphi) & \cos(\varphi) & 0 \\ 0 & 0 & 1 \end{bmatrix}, \tag{4}$$

$$R_{gNBy}(\theta) = \begin{bmatrix} \cos(\theta) & 0 & \sin(\theta) \\ 0 & 1 & 0 \\ -\sin(\theta) & 0 & \cos(\theta) \end{bmatrix}, \tag{5}$$

where $r = [r_x, r_y, r_z]^T$ is the gNB→UE direction vector in the gNB antenna array local CS; $R_{gNB}$ is a rotation matrix, that specifies the orientation of the gNB AA; $R_{gNBz}(\varphi)$ and $R_{gNBy}(\theta)$ are the components of the $R_{gNB}$ matrix, that define the rotation of the gNB antenna array in azimuth and elevation, respectively; $\varphi_r$ is the gNB AA rotation angle in azimuth; $\theta_r$ is the gNB AA rotation angle in elevation (AA tilt).

The calculation of the azimuth and elevation, yielding the direction of the gNB→UE, given the orientation of the gNB antenna array, is performed as:

$$\varphi_{gNB} = \tan^{-1}(r_y/r_x); \tag{6}$$

$$\theta_{gNB} = \tan^{-1}(r_z/d_{xy}); \tag{7}$$

$$d_{xy} = \sqrt{r_x^2 + r_y^2}. \tag{8}$$

Angles $\left[\varphi_{gNB}, \theta_{gNB}\right]$, at which the serving gNB sees UE (SOI link), are denoted as $\left[\varphi_{gNBs}, \theta_{gNBs}\right]$, and angles at which the neighboring gNB sees UE (SNOI link) are denoted as $\left[\varphi_{gNBi}, \theta_{gNBi}\right]$. The angles $\varphi_{gNBs}/\varphi_{gNBi}$ and $\theta_{gNBs}/\varphi_{gNBi}$ are used to calculate AA gain of the serving (SOI) and neighbor (SNOI) gNB towards the UE. The vector of the amplitude-phase distribution of signals (in the format of the vector of complex numbers) for antenna array elements, depending on the AOA/AOD of the signal, is calculated as:

$$s_a(\varphi, \theta) = e^{-j2\pi f \tau(\varphi, \theta)}; \tag{9}$$

$$\tau(\varphi, \theta) = \frac{X_a k_a(\varphi, \theta)}{c}; \tag{10}$$

$$k_a(\varphi, \theta) = -\begin{bmatrix} \cos(\theta)\sin(\varphi) \\ \cos(\theta)\sin(\varphi) \\ \sin(\theta) \end{bmatrix}; \tag{11}$$

$$X_a = \begin{bmatrix} x_{a1} & y_{a1} & z_{a1} \\ \vdots & \vdots & \vdots \\ x_{aN} & y_{aN} & z_{aN} \end{bmatrix}; \tag{12}$$

where $\varphi$ is the azimuth angle of arrival/departure; $\theta$ is the elevation angle of arrival/departure; $X_a$ is the array of coordinates $[N \times 3]$ of the AE ($N$ is the number of AEs); $k_a$ is the steering vector, calculated based on the AOA $[\varphi, \theta]$; $\tau$ is the signal arrival delay for each AE from the $[\varphi, \theta]$ direction. Formulas (9)–(12) are also used to calculate AA steering vector $w_a$, when substituting the necessary gNB→UE direction angles into them according to (6) and (7). Denote SOI gNB AA steering vector of coefficients $w_{as} = s_a(\varphi_{eNBs}, \theta_{eNBs})$, and SNOI gNB AA steering vector of coefficients $w_{as\_} = s_a(\varphi_{eNBs\_}, \theta_{eNBs\_})$. In this case, $\left[\varphi_{gNBs\_}, \theta_{gNBs\_}\right]$ are the angles of the direction of the neighboring gNB to its UE. The AA gain coefficient in a given direction $[\varphi, \theta]$, taking into account antenna array vector of steering coefficients $w_a$, is calculated as:

$$g(\varphi, \theta) = |w_a s_a(\varphi, \theta)|^2. \tag{13}$$

According to the formula (13), AA gain of the serving (SOI) gNB $g_s$ and AA gain of the neighboring (SNOI) gNB $g_i$ in the direction of UE is calculated, based on the angles $\left[\varphi_{gNBs}, \theta_{gNBs}\right]$ and $\left[\varphi_{gNBi}, \theta_{gNBi}\right]$, and vectors of steering coefficients $w_{as}$ and $w_{as\_}$ as:

$$g_s(\varphi_{gNBs}, \theta_{gNBs}) = |w_{as} s_a(\varphi_{gNBs}, \theta_{gNBs})|^2; \tag{14}$$

$$g_i(\varphi_{gNBi}, \theta_{gNBi}) = |w_{as\_} s_a(\varphi_{gNBi}, \theta_{gNBi})|^2. \tag{15}$$

Power (in dB), received by the UE from gNB, given $w_a$ vector, is calculated as:

$$P(g(\varphi, \theta), d) = 10 \log_{10}(g(\varphi, \theta)) - 20 \log_{10}\left(\frac{4\pi d}{\lambda}\right). \tag{16}$$

where $d$ is the distance between the UE and the gNB; $\lambda$ is the wavelength. In formula (16), the second term describes FSPL. The value of SIR for the UE in dB is calculated as:

$$SIR = P(g_s, d_s) - P(g_i, d_i), \tag{17}$$

where $g_s$ and $g_i$ denote AA gains of the serving (SOI) and neighboring (SNOI) gNBs in the direction of the UE, respectively; $d_s$ and $d_i$ are the distances from the UE to the serving and neighbor gNBs, respectively.

If there is an AA at the UE, its gain is calculated in a similar manner to the gNB. The vector that specifies the direction UE→gNB is calculated by Formula (1), but the resulting

vector is taken with the opposite sign $k_{ue} = -k$. Formulas (2)–(5) are used to translate the direction vector $k_{ue}$ into the UE AA local coordinate system. Formulas (6)–(8) are used to calculate the UE→gNB direction angles. The calculation of the amplitude-phase distribution vector and the vector of UE antenna array steering coefficients is performed by (9)–(12). UE AA gain in the direction of gNB is calculated by (13). The power, received by the UE with the AA, is calculated by (16) with an additional term $g_{UE}$, being the UE AA gain:

$$P\big(g_{gNB}, g_{UE}, d\big) = 10\log_{10}(g_{gNB}) + 10\log_{10}(g_{UE}) - 20\log_{10}\left(\frac{4\pi d}{\lambda}\right). \tag{18}$$

The SIR value for the UE with AA is calculated by (17), taking into account (18). The results of SIR calculations are shown in Figures 1 and 2 for the case of omnidirectional UE and $4 \times 4$ URA at gNB.

### 3.1.3. Evaluation of SIR Dependence on the Terrestrial Separation

The map of SIR values has the local maxima, the number and position of which is determined by the shape of the antenna array pattern. Peaks occur, when the AA pattern of the serving gNB (SOI) is directed with maximum gain at the UE and the neighbor gNB (SNOI) is directed with minimum gain at the UE. The SIR values decrease as the UE moves away from the serving gNB and approaches the neighboring gNB. At the same time, there is no dependence of the SIR values on the selected terrestrial separation $\Delta y$ along y-axis.

### 3.2. SIR Dependence on the Angular Separation of Two gNB-UE Links in 5G UDN with LAB

#### 3.2.1. Simulation Model for SIR Dependence on the Angular Separation

The model is designed to estimate the SIR dependence for the scenario of two gNB–UE radio links of 5G UDN with one serving gNB, using an AA for directional transmission. The structure of this simulation model is similar to that, described in Section 3.1. Simulation scenario assumes, that one gNB simultaneously serves two UEs by steering two independent ARP beams. The UEs are stationary, and their relative position is determined by two parameters: distance $d$ from the UE to the gNB and angular separation $\alpha$ between two UEs. This scenario does not use antenna array at the UE.

Figure 3a illustrates the simulation scenario of SIR dependence evaluation on angular separation $\alpha$ between two UEs of two gNB–UE links in 5G UDN with LAB. Figure 3b illustrates the map of SIR values in dB: x-axis shows values of distance $d$ from UE to gNB and y-axis shows values of angular separation $\alpha$ between two UEs. All dependences are calculated for $4 \times 4$ URA at gNB. Figure 3c shows the map of SIR values, that exceed a given threshold 15 dB; areas of the SIR map with values, exceeding 15 dB are shaded in green. Figure 3d shows the dependence of SIR on the values of the angular separation $\alpha$ for a given range $d$ from the UE to gNB.

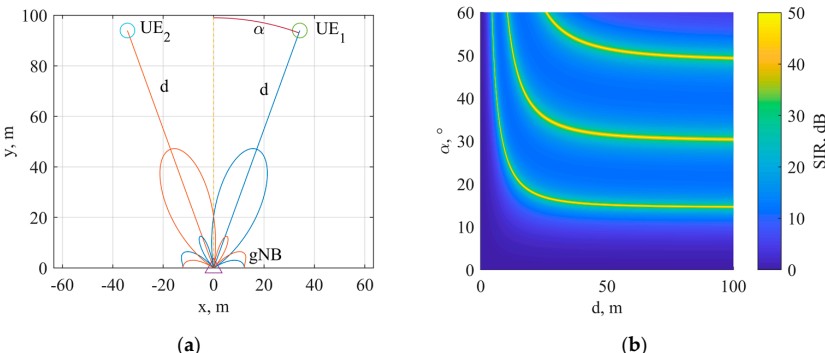

(a)                                                                                      (b)

**Figure 3.** *Cont.*

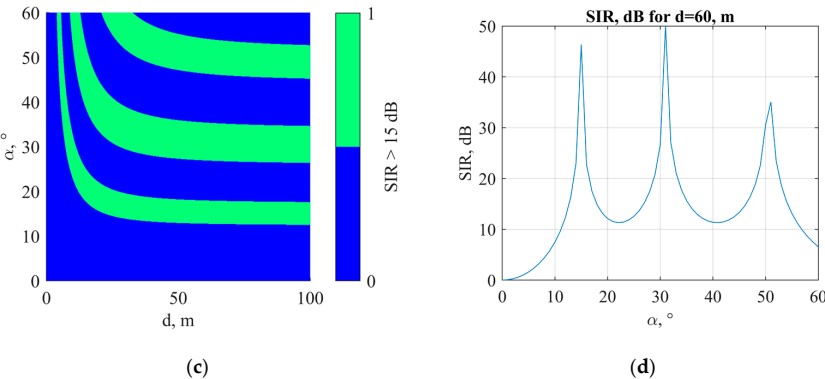

Figure 3. Simulation scenario 3: (**a**) gNB with two AA patterns and two UEs on the x-y plane; (**b**) Map of UE SIR values for a set of terrestrial and angular separation; (**c**) Map of SIR values exceeding a given threshold for a set of terrestrial and angular separation; (**d**) Map of SIR values for a set of angular separation and fixed terrestrial separation.

To create the AA, gNB and UE parameter structure, we use the createAnt, createNB and createUE functions, as described in Section 3.1.

### 3.2.2. Mathematical Model for SIR Dependence on the Angular Separation

The calculation of the signal-to-interference ratio in the downlink is performed for each pair of values from the set of parameters $d$ and $\alpha$. The SIR is calculated as the ratio of the power, received by the UE, being SOI, from gNB, transmitting in beamforming mode in its direction, to the power, received by this UE, from gNB, transmitting in beamforming mode in the neighboring UE direction. In this model, the SIR value reflects the ratio of the AA gains of two beams, formed by one gNB with antenna array in two given directions. Thus, SIR does not take into account FSPL, since according to scenario, both UEs are at the same distance $d$ from the gNB.

The matching of the coordinate systems of the UE and gNB AA is performed according to (1)–(5). The calculation of the azimuth and elevation angle, that define the gNB→UE direction, is performed using (6)–(8). The angles, at which gNB sees the UE, are denoted as $[\varphi_{gNBs}, \theta_{gNBs}]$. The calculation is carried out for one UE, let's designate it as UE$_S$.

Calculation of the AA gain in a given direction $[\varphi, \theta]$, taking into account the vector of AA steering coefficients $w_a$ in (9)–(12) and the antenna array element amplitude-phase distribution vector $s_a$ in (9)–(12), is performed according to (13). From (13), gNB AA beam gain $g_s$ for UE$_S$ and gain $g_i$ for neighboring UE in the UE$_S$ direction are calculated as:

$$g_s(\varphi_{eNBs}, \theta_{eNBs}) = |w_{as} s_a(\varphi_{eNBs}, \theta_{eNBs})|^2; \tag{19}$$

$$g_i(\varphi_{eNBs}, \theta_{eNBs}) = |w_{as\_} s_a(\varphi_{eNBs}, \theta_{eNBs})|^2; \tag{20}$$

where $w_{as}$ is the vector of steering coefficients for the gNB AA beam pattern, directed to UE$_S$; $w_{as\_}$ is a vector of steering coefficients for gNB AA beam pattern, directed to the neighboring UE. The signal-to-interference ratio for UE$_S$, expressed in dB, is defined as:

$$SIR = 10 \log_{10}(g_s(\varphi, \theta)) - 10 \log_{10}(g_i(\varphi, \theta)). \tag{21}$$

### 3.2.3. Evaluation of SIR Dependence on the Angular Separation

From Figure 3b it can be seen, that for ranges greater than 30 m, a repeating pattern of SIR values is observed; local maxima are determined by the antenna array pattern and indicate, that for given values of $\alpha$ and $d$, gNB beam, directed with maximum AA gain to the neighboring UE, is directed with minimum AA gain to the UE$_S$. For values of $d$, greater than 40 m, the pattern of SIR values stops to depend on $d$, since the tilt angle at long ranges changes slowly. There is also an area of low SIR values at $\alpha$ values less than 10 degrees,

regardless of the *d* values. The width of this region along the angle $\alpha$ is determined by the HPBW of the main lobe of the AA pattern (see Figure 4). From Figure 3c,d it can be seen, that SIR regions, exceeding specified threshold 15 dB, are repeated with a change in $\alpha$, that is determined by the shape of the AA pattern, in particular, by the number of side minima and maxima of the AA pattern side lobes, which in turn depends on the number of AA elements.

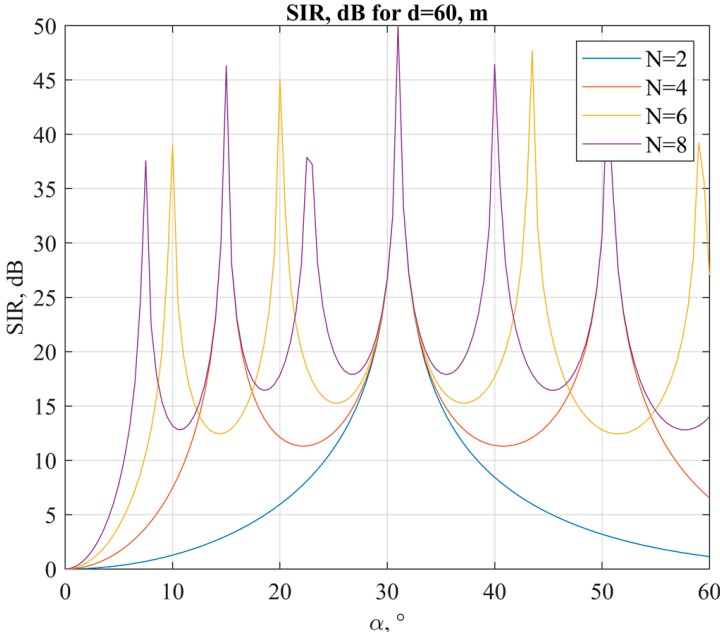

**Figure 4.** Dependence of SIR on angular separation at a given range.

Figure 4 shows SIR dependence on the angular separation $\alpha$ for given range $d = 60$ m with URA: if *N* is the number of AE on one URA side, the total number of AEs is $N^2$. From Figure 4 it can be seen, that with an increase in the number of AEs, the number of local SIR maxima increases in a given range of $\alpha$. The maxima are evenly spaced in the $\alpha$ range with a slight increase in the spacing between them, as $\alpha$ increases; the number of local maxima is equal to $N - 1$. With an increase in the number of AEs we can observe an increase in the minimum (drops in the SIR values) and, accordingly, the average SIR values in a given $\alpha$ range. The region of low SIR at the beginning of $\alpha$ range decreases with an increase in the number of AEs, which is caused by a decrease in the beam width of the main lobe of the AA pattern. For a threshold value of SIR = 10 dB, the width of the low SIR region is about 23, 11, 7, 5 degrees for the number *N* equal to 2, 4, 6, 8, respectively. For a URA there is a linear relationship between the increase in the number *N* and decrease in the angular width of the region of low SIR: increasing *N* by *k* times reduces the angular width by *k* times. The width of this region can serve as a criterion for the minimum allowable UE angular separation, when working with an array with a given number of AEs. A decrease in SIR is also observed, when $\alpha$ tends to 90 degrees, which is caused by an increase in the level of the AA side lobes at large deviation angles of the maximum of the URA beam pattern. This decrease in SIR can serve as a criterion for the maximum allowable angular separation, which is about 38, 57, 65, 70 degrees for the number *N* equal to 2, 4, 6, 8, respectively, with a threshold value of SIR = 10 dB. Similar conclusions are also valid for ULA with the number of AEs equal to *N*, when operating at ranges, at which the elevation angle changes slightly.

### 3.3. Transition from the Link to System Level Simulation Model of 5G UDN with LAB

According to the project "Location aware beamforming in mm-wave band ultra-dense radio access networks" [55], considered above link level simulation model of two links is a basis for ongoing research and system level simulation model development on the further

levels of abstraction with a set of links of one cell and a set of UDN cells with account of interference, coming from the neighboring cells.

In contrast to known private scenarios of LAB [28–54], proposed methodology for LAB in mmWave UDN for 5G and B5G, should take into account a set of conditions for networks development and scenarios for the device operation, including: (a) combining various primary measurements of a heterogeneous radio access network, as well as the accuracy of VUE location estimation and the speed of VUE motion; (b) configuration of antenna arrays of stationary gNB and mobile VUE devices; (c) the accuracy and speed of BF and determining the angle of arrival/departure (AOA/AOD); (d) geographical extent and density of gNB and UE. Consider briefly a set of conditions for networks development and scenarios for the device operation, mentioned above, with references to recent author investigations. Simulation model for vehicles tracking in 5G UDN, using combining range and bearing measurements in [23], shows the relationship between the accuracy of VUE location estimation, the speed of VUE motion, primary measurement accuracy and time interval. An open-source simulator for configuration of antenna arrays for gNB, tracking mobile VUE with location-aware beamforming for the case of two links, described above, is available at [76]. Correspondence between the accuracy and speed of BF and AOA/AOD estimation, is investigated in [24] and [27] respectively. Grid model for simulation of geographical extent and density of gNB and UE is proposed in [18].

On the link level the relationship between the speed of VUE motion $v$ and primary measurement time interval $T$ for VUE location estimation is defined by the discrete step $\Delta x$ of VUE linear trajectory, parallel to x-axis, in the createScenario function inside LAB link level simulator [76] in following order:

$$\Delta x = v \cdot T. \tag{22}$$

During this discrete step developed simulation model performs SIR calculation according to Formulas (1)–(21). The vector, specifying gNB→UE direction in the global CS, is calculated according to (1) and accounts UE coordinates $x_{UE}$ during further location-aware beamforming procedures. VUE positioning inaccuracy in the simulation model can be accounted by adding some random error to $x_{UE}$ on each discrete step $\Delta x$ during VUE motion according to its trajectory. For example, 3 m VUE location estimation accuracy is achieved for 60 km/h VUE if at least two gNB combine range and bearing measurements with 10 ns and 12° error respectively every 0.01 s or one NR radio frame [23]; to achieve sub-meter positioning accuracy additional measurements must be collected. Thus, proposed methodology for LAB in 5G UDN takes into account mentioned above set of conditions for networks development and scenarios for the device operation on the link level.

On the system level to expand the proposed method for scenarios of more links we can utilize dense urban grid model [18], according to which base stations are placed with predefined ISD between neighboring gNB. System level simulation model development with a set of links of one gNB and a set of gNBs is the object of future research.

## 4. Conclusions

The contribution of current research is threefold. First, it substantiates location-aware beamforming for efficient maintenance of affordable level of interference in space-division multiple access scenarios, based on the state-of-the art analysis. Second, it presents and describes publicly released open-source simulator for signal-to-interference ratio calculation with two location-aware beamforming links. Third, it substantiates allowable UE density by the criterion of their terrestrial and angular separation. For considered urban micro-cell scenario it demonstrates insignificant SIR dependence on UE terrestrial separation and significant SIR dependence on UE angular separation. In particular, the width of the low SIR region decreases with the antenna array size increase, thus the width of this region can serve as a criterion for the minimum allowable UE angular separation in 5G UDN scenarios, when working with an array of given size. The matlab source code for presented

simulation model can be downloaded at: https://github.com/grihafokin/LAB_link_level (accessed on 20 October 2022).

**Author Contributions:** Conceptualization, G.F.; methodology, G.F. and D.V.; software, D.V.; validation, D.V.; formal analysis, G.F.; investigation, G.F. and D.V.; resources, G.F.; data curation, G.F.; writing—original draft preparation, G.F.; writing—review and editing, G.F.; visualization, D.V.; supervision, G.F.; project administration, G.F.; funding acquisition, G.F. All authors have read and agreed to the published version of the manuscript.

**Funding:** This research was funded by the Russian Science Foundation Grant No. 22-29-00528, https://rscf.ru/project/22-29-00528/ (accessed on 8 November 2022).

**Data Availability Statement:** Not applicable.

**Conflicts of Interest:** The authors declare no conflict of interest.

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
