# Peer review of "Model for Interference Evaluation in 5G Millimeter-Wave Ultra-Dense Network with Location-Aware Beamforming"

_information, doi:10.3390/info14010040_

Round 1

Reviewer 1 Report

The authors propose an open-source simulator to evaluate the interference in 5G mmWave UDN with LAB, which substantiates the allowable UE density numerically. Some comments from this reviewer are listed as follows.

1.  It is suggested that the authors could provide more detailed analysis and theory of the mathematical analysis of the proposed simulation model.

2.  How are the various primary measurements of heterogeneous radio access network combined in the simulated scenarios?

3. What is the main difference between the proposed methodology for LAB and the existing ones [24-50] ?

4.  The impact of terrestrial and angular separation on the SIR dependence is analyzed in Section 3.1 and 3.2, respectively. However, the terrestrial and angular separation are not the only aspects that influence the maximum allowable UE density. Other aspects, e.g., the modulation types considered in the base station and the user equipment, the antenna configuration for the transceiver, may also contribute to the calculation of the maximum allowable UE density.  Thus, the considered scenario in this manuscript may be oversimplified. Moreover, the substantiation of the allowable UE density is vague, and should be further explained.

 5. The English of this manuscript should be revised properly. 

Author Response

Response to Reviewer 1

The authors propose an open-source simulator to evaluate the interference in 5G mmWave UDN with LAB, which substantiates the allowable UE density numerically. Some comments from this reviewer are listed as follows. Comments from Reviewer 1 are listed as follows:

  1. It is suggested that the authors could provide more detailed analysis and theory of the mathematical analysis of the proposed simulation model.

Thank you for your comment. We fully agree, that more detailed analysis and theory of the mathematical analysis of the proposed simulation model is required. Such description is presented in our first work [1] Davydov, V.; Fokin, G.; Moroz, A.; Lazarev, V. (2022). Instantaneous Interference Evaluation Model for Smart Antennas in 5G Ultra-Dense Networks. In Proceedings of the International Conference on Next Generation Wired/Wireless Networking NEW2AN ruSMART 2021. Lecture Notes in Computer Science 2021, 13158, 365–376. https://doi.org/10.1007/978-3-030-97777-1_31. However, that investigation considered single scenario and utilized built in functions of Mathwork Phased Array System Toolbox. Presented in the current work open-source simulator utilizes only primitive functions, which are described by formulas (1)-(21) and can be verified by:

 https://github.com/grihafokin/LAB_link_level

Added paragraph in the beginning of the Section 3.

It should be noted, that general structure of presented link level simulation model was already detailed in [1]. However, investigation [1] utilized built in functions of MathWorks Phased Array System Toolbox [71]. Presented in the current work open-source simulator utilizes only primitive MathWorks functions and is available at [72].

  1. How are the various primary measurements of heterogeneous radio access network combined in the simulated scenarios?

Thank you for your comment. Explanation is in the end of the Section 2.

Combination of various primary measurements of heterogeneous radio access network for UE location estimation and further beamforming in the simulated scenarios can be accounted with positioning inaccuracy, defined by standard deviation of UE coordinates calculation. Further simulation results are performed for the case of perfect positioning accuracy, and even for such scenario it demonstrates considerable SIR fluctuation, that needs to be accounted for, when assessing UE terrestrial and angular separation.

  1. What is the main difference between the proposed methodology for LAB and the existing ones [24-50]?

Thank you for your comment. Main difference between the proposed methodology for LAB and the existing [24-50] were stated in the following paragraphs of Introduction:

Initial research in the field of LAB has already been carried out over the past years, but mainly for a particular scenario of one radio link between a base station and a UE in order to estimate UE location [24–27] and for a particular scenario of the mutual influence of two links between one base station and two fixed UE for the purpose of interference evaluation [28–34]. The problem of evaluating interference in radio links with beamforming for vehicles is complicated by following factors: significant (tens of dB) dependence of the instantaneous signal-to-noise-plus interference ratio (SINR) on the current spatial [1] and angular [16] separation of the VUE; interdependence of channel coherence time and beamwidth in vehicular channels [35–37]; dependence of beamformed radio link capacity on UE positioning uncertainty [38]; employed beam tracking approaches [39, 40]; beam alignment techniques [41–44] and location-aided channel estimation [45–49].

One of the first works, concerning LAB for fixed UE [50], revealed, that using position enables in creating transmit precoding BF vector for line-of-sight (LOS) scenarios without signaling overhead. In particular, proposed approach was to limit beamforming to spatial area, bounded by angular sector with inaccuracy of the GPS positioning up to 20 m.

Main idea and novelty of proposed model is explained in the end of Section 2.

Presented background reveals several problems in organizing directional links in 5G UDN and substantiates location-aware beamforming as a solution. Consider further briefly approaches for interference evaluation in 5G mmWave UDN.

Investigation [70] consider deployment of mmWave gNB microcells in the areas with high UE density in addition to traditional sub-6 GHz macrocells. While sub-6 GHz Base Stations (BS) and UE are equipped with omni-directional antennas, mmWave gNB utilize directional transmission. Using stochastic geometry and assumption of UE distribution according to Poisson cluster process, authors in [70] develop comprehensive system level analytical framework and analyze various performance metrics, including interference characterization. Monte Carlo simulation validate analytical results and prove significant network performance improvement with deployment of mmWave gNB with directional beamforming. However, on the link level authors in [70] consider directional transmission with model of sectored antenna pattern. According to such model antenna gain can take only two values: gs, if the UE is within gNB main beamwidth and g's, if the UE is outside gNB main beamwidth, where gs and g's denote the gains of the main lobe and side lobe respectively. Interference characterization would be more accurate if we account complex radiation pattern for antenna array types, depending on the number of array elements.

More realistic approach for interference evaluation in 5G mmWave UDN is reported in [28–34], however, for stationary UE cases. To the best of our knowledge, the problem of interference evaluation for the case of mobile UE had not yet been thoroughly investigated. In this work we consider the problem of interference evaluation for the case of two gNB-UE links with mobile UE, as a basis for further evaluation on the level of a set of links inside single UDN microcell and finally, on the network level of a set of UDN microcells. Despite the fact, that considered case of interference evaluation includes only two gNB-UE links, it is rather challenging. First challenge consists in the instantaneous dependency of signal to interference ratio on UE angular and spatial separation, which changes during their mutual motion. Second challenge includes strong SIR dependence on the nonlinear ARP, defined by the AA type and the number of array elements. Third challenge is due to the need to take into account UE positioning accuracy for gNB, utilizing location-aware beamforming. Oversimplification of these factors can lead to the network SIR overestimation or underestimation, which can take values of several tens of dB.

The novelty of current work is in instantaneous SIR evaluation model for cases of two links between two mobile UE and two stationary gNB, equipped with smart antennas, which perform location-aware beamforming during UE motion in predefined scenarios, accounting for their SOI and SNOI roles. Developed and publicly released open-source simulator includes thorough account of UE mobility and complex models for ARP for various AA types and the number of array elements. Further simulation results demonstrate considerable SIR fluctuation, that needs to be accounted for, when assessing UE terrestrial and angular separation.  In the further subsection we will consider simulation model of two gNB-UE radio links.

  1. The impact of terrestrial and angular separation on the SIR dependence is analyzed in Section 3.1 and 3.2, respectively. However, the terrestrial and angular separation are not the only aspects that influence the maximum allowable UE density. Other aspects, e.g., the modulation types considered in the base station and the user equipment, the antenna configuration for the transceiver, may also contribute to the calculation of the maximum allowable UE density. Thus, the considered scenario in this manuscript may be oversimplified. Moreover, the substantiation of the allowable UE density is vague, and should be further explained.

Thank you for your comment. We fully agree with the fact, that terrestrial and angular separation are not the only aspects, that influence the maximum allowable UE density, and added one paragraph at the end of Section 3.1.1.

Terrestrial and angular separation are not the only aspects, influencing the maxi-mum allowable UE separation. From the physical layer perspective other aspects, such as utilized modulation and coding schemes at the gNB and UE, also contribute to the admissible SINR. However, from the perspective of link and network layer of abstraction, resulting map with SIR values, exceeding a given threshold, could serve as a reference point to assess allowable UE separation in terms of interference distance at first approximation. From Figure 1(d) and Figure 2(d) we can conclude, that at some point, defined by interference distance, UE1, being SOI, begin to experience SIR deterioration, when ARP from SOI gNB1 and SNOI gNB2 are directed to closely located UE1 and UE2 respectively. To over-come such SIR deterioration gNB inter-site distance (ISD) could be reduced.

Concerning “substantiation of the allowable UE density” we added a sentence with the following reference.

Besides interference distance, authors in [52] investigate BS density per km2 and km3 for allowable network densification evaluation and state, that directional transmissions are seen as one of the solutions to push fundamental densification limits as far as possible. 

As for the antenna configuration for the transceiver, contributed model accounts for most popular antenna array types: uniform linear array (ULA), uniform rectangular or planar array (URA) and uniform circular array (UCA). Presented results include URA at the gNB and omnidirectional antenna at the UE, as it was investigated in [1] Davydov, V.; Fokin, G.; Moroz, A.; Lazarev, V. (2022). Instantaneous Interference Evaluation Model for Smart Antennas in 5G Ultra-Dense Networks. In Proceedings of the International Conference on Next Generation Wired/Wireless Networking NEW2AN ruSMART 2021. Lecture Notes in Computer Science 2021, 13158, 365–376. https://doi.org/10.1007/978-3-030-97777-1_31

Concerning “oversimplified scenario” we would like to defend and justify it with the following arguments.

We fully agree with the fact, that 2 UE and 2 BS is somehow simplified scenario. However, in the introduction we mentioned, that “Fundamental purpose of the project “Location aware beamforming in mm-wave band ultra-dense radio access networks” [51] is to develop scientifically grounded methodology for ARP control, based, on the positioning of UE for scenarios of a separate link, two links, a set of links of one cell and a set of UDN cells.” Thus, considered model of 2 UE and 2 BS is a basis for ongoing research and model development on the further levels of abstraction with a set of links of one cell and a set of UDN cells, where we surely account interference, coming from the neighboring cells.

To defend and justify the need for modeling of 2 UE and 2 BS scenario we added additional substantiation in the end of the Section 2.

More realistic approach for interference evaluation in 5G mmWave UDN is reported in [28–34], however, for stationary UE cases. To the best of our knowledge, the problem of interference evaluation for the case of mobile UE had not yet been thoroughly investigated. In this work we consider the problem of interference evaluation for the case of two gNB-UE links with mobile UE, as a basis for further evaluation on the level of a set of links inside single UDN microcell and finally, on the network level of a set of UDN microcells. Despite the fact, that considered case of interference evaluation includes only two gNB-UE links, it is rather challenging. First challenge consists in the instantaneous dependency of signal to interference ratio on UE angular and spatial separation, which changes during their mutual motion. Second challenge includes strong SIR dependence on the nonlinear ARP, defined by the AA type and the number of array elements. Third challenge is due to the need to take into account UE positioning accuracy for gNB, utilizing location-aware beamforming. Oversimplification of these factors can lead to the network SIR overestimation or underestimation, which can take values of several tens of dB.

  1. The English of this manuscript should be revised properly.

Thank you for your comment. English of this manuscript has been revised with a native English speaker.

Reviewer 2 Report

This paper evaluates the system-level performance of an ultra-dense 5G millimeter wave network with location-aware beamforming. To avoid the time-consuming channel training therefore the UE location is utilized to enable LAB and SDMA. The system performance is evaluated in terms of received SIR. 

I have the following comment regarding the manuscript.

Major comments:

1) The quality of the figures can be improved in terms of dpi.

2) Better to use \arctan function in latex instead of text in equations 6 and 7 (check all manuscript)

3) line 443 gNB->UE what does it mean if it is an arrow define it properly (check all manuscript)

4) A very simplified scenario with 2 UE and 2 BS is considered in the simulation. Is not it an unrealistic network model while normally the interference may come from the neighboring cell more than 1? How can you defend such a simplified unrealistic model?

5) The performance evaluation of hybrid millimeter wave is already been published in the literature. Is the only novelty of this work is consider location-aware beamforming?

6) How can you compare this work with the recently published work in [1] by adding discussion in the introduction or conclusion/future work

[1] Ullah, Arif, et al. "Hybrid millimeter wave heterogeneous networks with spatially correlated user equipment." Digital Communications and Networks (2022).

Author Response

Response to Reviewer 2

This paper evaluates the system-level performance of an ultra-dense 5G millimeter wave network with location-aware beamforming. To avoid the time-consuming channel training therefore the UE location is utilized to enable LAB and SDMA. The system performance is evaluated in terms of received SIR.  I have the following comment regarding the manuscript. Major comments:

1) The quality of the figures can be improved in terms of dpi.

Thank you for your comment. All figures were improved up to 600 dpi.

2) Better to use \arctan function in latex instead of text in equations 6 and 7 (check all manuscript)

Thank you for your comment. \arctan function is replaced with   function from Microsoft Word Equation editor through all manuscript.

3) line 443 gNB->UE what does it mean if it is an arrow define it properly (check all manuscript)

Thank you for your comment. Symbol “->” means downlink transmission and is replaced with an arrow “→” through all manuscript.

4) A very simplified scenario with 2 UE and 2 BS is considered in the simulation. Is not it an unrealistic network model while normally the interference may come from the neighboring cell more than 1? How can you defend such a simplified unrealistic model?

Thank you for your comment. We fully agree with the fact, that 2 UE and 2 BS is somehow simplified scenario. However, in the introduction we mentioned, that “Fundamental purpose of the project “Location aware beamforming in mm-wave band ultra-dense radio access networks” [51] is to develop scientifically grounded methodology for ARP control, based, on the positioning of UE for scenarios of a separate link, two links, a set of links of one cell and a set of UDN cells. Thus, added a sentence in the end of Introduction:

Considered model of two links is a basis for ongoing research and model development on the further levels of abstraction with a set of links of one cell and a set of UDN cells with account of interference, coming from the neighboring cells.

To defend and justify the need for modeling of 2 UE and 2 BS scenario we added additional substantiation in the end of the Section 2.

More realistic approach for interference evaluation in 5G mmWave UDN is reported in [28–34], however, for stationary UE case. To the best of our knowledge, the problem of interference evaluation for the case of mobile UE had not yet been thoroughly investigated. In this work we consider the problem of interference evaluation for the case of two gNB-UE links with mobile UE, as a basis for further evaluation on the level of a set of links inside single UDN microcell and finally, on the network level of a set of UDN microcells. Despite the fact, that considered case of interference evaluation includes only two gNB-UE links, it is rather challenging. First challenge consists in the instantaneous dependency of signal to interference ratio on UE angular and spatial separation, which changes during their mutual motion. Second challenge includes strong SIR dependence on the nonlinear ARP, defined by the AA type and the number of array elements. Third challenge is due to the need to take into account UE positioning accuracy for gNB, utilizing location-aware beamforming. Oversimplification of these factors can lead to the network SIR overestimation or underestimation, which can take values of several tens of dB.

5) The performance evaluation of hybrid millimeter wave is already been published in the literature. Is the only novelty of this work is consider location-aware beamforming?

Thank you for your comment. Hybrid millimeter wave beamforming really had already been thoroughly investigated in the last decade, and this work doesn’t consider it. The novelty of this work is detailed in the end of the Section 2.

The novelty of current work is in instantaneous SIR evaluation model for cases of two links between two mobile UE and two stationary gNB, equipped with smart antennas, which perform location-aware beamforming during UE motion in predefined scenarios, accounting for their SOI and SNOI roles. Developed and publicly released open-source simulator includes thorough account of UE mobility and complex models for ARP for various AA types and the number of array elements. Further simulation results demonstrate considerable SIR fluctuation, that needs to be accounted for, when assessing UE terrestrial and angular separation.

6) How can you compare this work with the recently published work in [1] by adding discussion in the introduction or conclusion/future work: [1] Ullah, Arif, et al. "Hybrid millimeter wave heterogeneous networks with spatially correlated user equipment." Digital Communications and Networks (2022).

Thank you for your comment. Comparison is added in the end of Section 2.

Investigation [70] consider deployment of mmWave gNB microcells in the areas with high UE density in addition to traditional sub-6 GHz macrocells. While sub-6 GHz Base Stations (BS) and UE are equipped with omni-directional antennas, mmWave gNB utilize directional transmission. Using stochastic geometry and assumption of UE distribution according to Poisson cluster process, authors in [70] develop comprehensive system level analytical framework and analyze various performance metrics, including interference characterization. Monte Carlo simulation validate analytical results and prove significant network performance improvement with deployment of mmWave gNB with directional beamforming. However, on the link level authors in [70] consider directional transmission with model of sectored antenna pattern. According to such model antenna gain can take only two values: , if the UE is within gNB main beamwidth and , if the UE is outside gNB main beamwidth, where  and  denote the gains of the main lobe and side lobe respectively. Interference characterization would be more accurate if we account complex radiation pattern for antenna array types, depending on the number of array elements.

Round 2

Reviewer 1 Report

The manuscript has been revised, while there are still some suggestions to be addressed.

1. In the section of Introduction, the authors say that the proposed methodology for LAB in mmWave UDN for 5G and B5G should take in account a set of conditions, including the accuracy and speed of VUE. The expression “accuracy of VUE” is ambiguous, and it seems that the authors haven’t taken into account the speed of VUE in the manuscript.

2. In section 3, “the createAnt function” and “the createNB function” are mentioned. The authors are suggested to insert the footnote to simply introduce these two functions, e.g., whether the functions are provided within the open-source simulator itself.

3.  This work aims to evaluate interference in 5G mmWave UDN with LAB for scenarios of two links, I wonder whether the proposed method can be expanded for scenarios of more links. The authors could explain it simply.

4. In the manuscript, the stationary gNB and mobile UE are supposed to use the same type of antenna array, is the considered scenario reasonable? As far as I know, the base stations in 5G networks are usually equipped with larger antenna arrays compared with user equipments, and the type of antenna arrays may be not same.

5. In section 3, “The terrestrial separation of the UE trajectories is determined by the value of ?”, “the distance between AE ? =?⁄2”. “?” is defined as half wavelength, and how could ? determine the terrestrial separation of the UE trajectories ?

6. The name of two gNBs, gNBand gNB2, are suggested to be added in Figure 1.(a) and Figure 2.(a).

7. The references to 5G location architecture and technologies are inadequate. The following papers can be added as references. 

[R1] Prospective positioning architecture and technologies in 5G networks, DOI: 10.1109/MNET.2017.1700066.

[R2] A Survey of Enabling Technologies for Network Localization, Tracking, and Navigation, DOI: 10.1109/COMST.2018.2855063.

Author Response

Response to Reviewer 1

The manuscript has been revised, while there are still some suggestions to be addressed.:

  1. In the section of Introduction, the authors say that the proposed methodology for LAB in mmWave UDN for 5G and B5G should take in account a set of conditions, including the accuracy and speed of VUE. The expression “accuracy of VUE” is ambiguous, and it seems that the authors haven’t taken into account the speed of VUE in the manuscript.

Thank you for your comment. We agree, that the expression “accuracy of VUE” is ambiguous. Thus, it was reformulated inside the new subsection «3.3. Transition from the Link to System Level Simulation Model of 5G UDN with LAB»:

«… a) combining various primary measurements of a heterogeneous radio access network, as well as the accuracy of VUE location estimation and the speed of VUE motion; …».

To simply explain an approach of taking into account the speed of VUE and addressing suggestion 3, authors added new subsection «3.3. Transition from the Link to System Level Simulation Model of 5G UDN with LAB»:

According to the project “Location aware beamforming in mm-wave band ultra-dense radio access networks” [55], considered above link level simulation model of two links is a basis for ongoing research and system level simulation model development on the further levels of abstraction with a set of links of one cell and a set of UDN cells with account of interference, coming from the neighboring cells.

In contrast to known private scenarios of LAB [28–54], proposed methodology for LAB in mmWave UDN for 5G and B5G, should take into account a set of conditions for networks development and scenarios for the device operation, including: a) combining various primary measurements of a heterogeneous radio access network, as well as the accuracy of VUE location estimation and the speed of VUE motion; b) configuration of antenna arrays of stationary gNB and mobile VUE devices; c) the accuracy and speed of BF and determining the angle of arrival/departure (AOA/AOD); d) geographical extent and density of gNB and UE. Consider briefly a set of conditions for networks development and scenarios for the device operation, mentioned above, with references to recent author investigations. Simulation model for vehicles tracking in 5G UDN, using combining range and bearing measurements in [23], shows the relationship between the accuracy of VUE location estimation, the speed of VUE motion, primary measurement accuracy ant time interval. An open-source simulator for configuration of antenna arrays for gNB, tracking mobile VUE with location-aware beamforming for the case of two links, described above, is available at [76]. Correspondence between the accuracy and speed of BF and AOA/AOD estimation, is investigated in [24] and [27] respectively. Grid model for simulation of geographical extent and density of gNB and UE is proposed in [18].

On the link level the relationship between the speed of VUE motion  and primary measurement time interval  for VUE location estimation is defined by the step , corresponding to discrete step VUE linear trajectory, parallel to x-axis, in the following order:

During this discrete step developed simulation model performs SIR calculation according to formulas (1)–(21). The vector, specifying gNB→UE direction in the global CS, is calculated according to (1) and accounts UE coordinates  during further location-aware beamforming procedures. VUE positioning inaccuracy in the simulation model can be accounted by adding some random error to  on each discrete step  during VUE motion according to its trajectory. For example, 3m VUE location estimation accuracy is achieved for 60 km/h VUE if at least two gNB combine range and bearing measurements with 10 ns and 12° error respectively every 0.01 s or one NR radio frame [23]; to achieve sub-meter positioning accuracy additional measurements must be collected. Thus, proposed methodology for LAB in 5G UDN takes into account mentioned above set of conditions for networks development and scenarios for the device operation on the link level.

  1. In section 3, “the createAnt function” and “the createNB function” are mentioned. The authors are suggested to insert the footnote to simply introduce these two functions, e.g., whether the functions are provided within the open-source simulator itself.

Thank you for your comment. In addition to the reference in the beginning of Section 3:

«Presented in the current work open-source simulator utilizes only primitive MathWorks functions and is available at [76]»

the footnote is also inserted to the createAnt function in the section 3.1.1:

  All functions, comprising open-source simulator, are available at https://github.com/grihafokin/LAB_link_level

  1. This work aims to evaluate interference in 5G mmWave UDN with LAB for scenarios of two links, I wonder whether the proposed method can be expanded for scenarios of more links. The authors could explain it simply.

Thank you for your comment. To simply explain an approach of expanding scenario of two links on the scenario with more links, authors added additional paragraph with reference to [18]

https://ieeexplore.ieee.org/abstract/document/9078650

inside the new subsection «3.3. Transition from the Link to System Level Simulation Model of 5G UDN with LAB»:

On the system level to expand the proposed method for scenarios of more links we can utilize dense urban grid model [18], according to which base stations are placed with predefined ISD between neighboring gNB. System level simulation model development with a set of links of one gNB and a set of gNBs is the object of future research.

  1. In the manuscript, the stationary gNB and mobile UE are supposed to use the same type of antenna array, is the considered scenario reasonable? As far as I know, the base stations in 5G networks are usually equipped with larger antenna arrays compared with user equipments, and the type of antenna arrays may be not same.

Thank you for your comment. We fully agree with the fact, that base stations in 5G networks are usually equipped with larger antenna arrays, compared with user equipment. Presented results in section 3.1 are for the case of omni-directional UE and 4×4 URA at gNB, which is proved by the sentence in the end of subsection 3.1.2:

«The results of SIR calculations are shown in Figure 1 and Figure 2 for the case of omni-directional UE and 4×4 URA at gNB.»

Misleading sentence in the subsection 3.1.1

«Stationary gNB and mobile UE use same type of AA.»

was refined to

« Stationary gNB and mobile UE can use same type of AA, however, gNB in 5G networks are usually equipped with larger AA, compared with UE.»

  1. In section 3, “The terrestrial separation of the UE trajectories is determined by the value of ?”, “the distance between AE ? =?⁄2”. “?” is defined as half wavelength, and how could ? determine the terrestrial separation of the UE trajectories ?

Thank you for your comment. This is really my mistake.

Misleading part in sentence in the subsection 3.1.1:

«…the distance between AE , which is chosen to be equal to the half wavelength …»

was refined to

«…the distance between AE is  and is chosen to be equal to the half wavelength …»

to exclude the same notation of , which is the distance between the UE and the gNB throughout the whole link level model in section 3.1 and 3.2,

misleading part in sentence in the subsection 3.1.1:

The terrestrial separation of the UE trajectories is determined by the value of .

was refined to

The terrestrial separation of the UE linear trajectories, parallel to the x-axis, are depicted in Figure 1 and are determined by the value  along y-axis, varying from 0 m to 10 m (street width) with predefined step . 

  1. The name of two gNBs, gNB1 and gNB2, are suggested to be added in Figure 1.(a) and Figure 2.(a).

Thank you for your comment. Names of two gNBs, gNB1 and gNB2, were originally plotted in Figure 1.(a) and Figure 2.(a), however, due to antenna radiation pattern, their notations were shadowed. Now it's fixed.

  1. The references to 5G location architecture and technologies are inadequate. The following papers can be added as references.

[R1] Prospective positioning architecture and technologies in 5G networks, DOI: 10.1109/MNET.2017.1700066.

[R2] A Survey of Enabling Technologies for Network Localization, Tracking, and Navigation, DOI: 10.1109/COMST.2018.2855063.

Thank you for your comment. These two papers are really well-known works and were added as references [20] and [21] in the following paragraph of Introduction on page 2:

A survey of enabling technologies for network localization, tracking, and navigation in [20] highlights shift from positioning in two dimensions to three-dimensional location estimation in 5G UDN environments using range-free localization schemes. Overview of positioning architectures in previous generations of cellular networks and prospective positioning architecture and technologies in [21] discuss potential in achieving sub-meter accuracy in 5G networks.

We would also like to point out, that these two well-known works were already studied and referenced in authors earlier published investigations. For example,

in 2020 work by V. O. Lazarev and G. A. Fokin, "Positioning Performance Requirements Evaluation for Grid Model in Ultra-Dense Network Scenario," 2020 Systems of Signals Generating and Processing in the Field of on Board Communications, 2020, pp. 1-6, doi: 10.1109/IEEECONF48371.2020.9078650 these two well-known works are referenced as [10] and [11]:

https://ieeexplore.ieee.org/document/9078650/references#citations

          in 2021 work by G. Fokin and V. Sevidov, "Topology Search Using Dilution of Precision Criterion for Enhanced 5G Positioning Service Area," 2021 13th International Congress on Ultra Modern Telecommunications and Control Systems and Workshops (ICUMT), 2021, pp. 131-136, doi: 10.1109/ICUMT54235.2021.9631679 these two well-known works are referenced as [18] and [19].

https://ieeexplore.ieee.org/document/9631679/references#references

Reviewer 2 Report

I have no further comments

Author Response

Thank you for your feedback.
